# InfoMax-based Resampling for Dataset Balance and Diversity

## Abstract

We propose a principled reweighting framework that moves empirical data toward uniform coverage through implicit differential entropy maximization. The core idea replaces intractable entropy maximization with a mutual information proxy and derives variational estimators under change of measure, yielding a consistent, low-variance weighted InfoNCE-based objective. Learned weights are immediately usable for data filtration and imbalance-aware sampling.

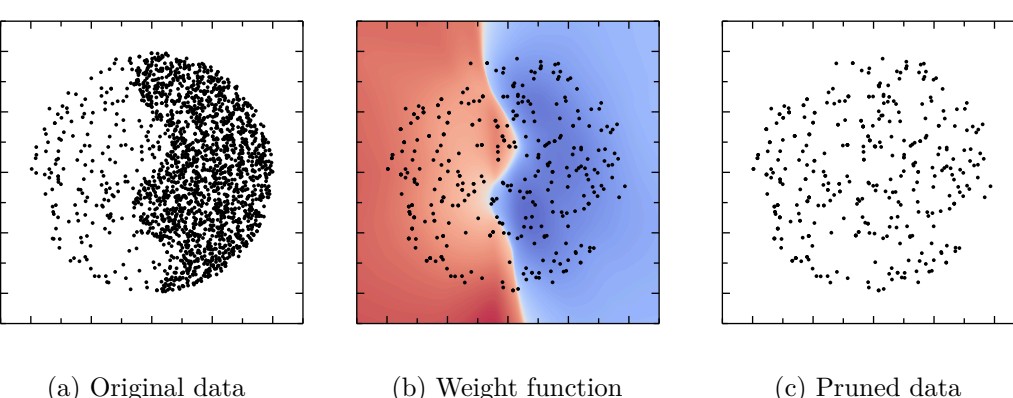

(a) Original data       (b) Weight function       (c) Pruned data

## 1 Introduction

The success of modern deep neural networks is largely attributed to large-scale datasets, which have driven many recent breakthroughs in Deep Learning (Brown et al., 2020; LeCun et al., 2015; Radford et al., 2021). Yet, training on such massive collections, from benchmark image sets to web-scale corpora brings substantial storage and computational costs and amplifies data redundancy (Kaplan et al., 2020). This has sparked interest in methods that reduce dataset size through distillation, pruning, and filtration, so that models can be stored and trained more efficiently without losing essential information.

The initial necessity for these excessively large datasets often comes from biases and imbalances inherent in data scraping: due to modern datasets being mostly collected from uncurated and unrefined sources, there is a fundamental misalignment between data generation and data collection (Brown et al., 2024; Chen et al., 2024; Huang et al., 2024; Roh et al., 2021; Vargas et al., 2023). Typical datasets cover the same manifolds as corresponding true data-generating distributions, yet they are often unrepresentative and biased in their allocation of probability mass. Some regions are heavily oversampled because they are easy to collect, while other, potentially informative regions are rare or hard to obtain. Ultimately, this misalignment yields low-diversity and high-redundancy data. This is especially critical for training Deep Learning models which are already biased toward low-diversity distributions, e.g., transformers (Ren and Liu, 2025).

Information theory suggests that the ability to compress or prune such datasets while not degrading downstream performance can be attributed to oversampled data being low-entropy (Cover and Thomas, 2006; Polyanskiy and Wu, 2024; Shannon, 1948). Conversely,

entropy maximization is known to balance and diversify datasets (Butakov et al., 2025; Li et al., 2020). Motivated by this, we aim to produce a principled reweighting of the distribution so that underrepresented regions receive appropriate emphasis, without requiring explicit estimation of the manifold or of high-dimensional densities. In particular, we propose a model-free and computationally efficient method based on information-theoretic principles to perform a uniformization, i.e., aligning a given data distribution so that its mass is spread more evenly over the data manifold.

This perspective differs from standard compression-oriented approaches. Distillation and coreset selection typically define objectives tied to model performance or to preserving specific statistical properties of the distribution, and they usually return either a synthetic tiny set or a selected subset of real examples. By contrast, our method operates at the level of probability measures and provides a weight function (density ratio) defined over the original data. Those weights can be used directly for imbalance-aware sampling, filtering, or to build coresets, but obtaining them does not require training task-specific models or generating synthetic points. Thus our method is more general in scope as it targets distribution alignment first and then supplies a tool that can support downstream compression or pruning when desired.

In summary, our main contributions are the following:

1. Motivated by information theory, we propose a novel approach to dataset balancing through distribution uniformization.

2. This task is then reframed as variational mutual information estimation under a change of measure, which allows one to avoid explicit density or support estimation.

3. We derive a weighted based objective; optimizing this objective produces weights that can be used for sampling, dataset balancing, and other downstream compression tasks.

4. We validate through synthetic and real-data diagnostics that our learned importance weights produce more uniform class distributions and improve instance-level coverage, as measured by standard deviation-from-uniform and coverage/diversity metrics.

We discuss the related works in Section 2 and provide the necessary background from information theory in Section 3. We then derive our method in Section 4 with theoretical bounds that justify the proxy objective. In Section 5, an experimental evaluation of the proposed approach is provided. Finally, we discuss our results in Section 6. Complete proofs are provided in Section A. Implementation details can be found in Section 5.1 and Section B.

## 2 BACKGROUND

In this section, we briefly review three main lines of related prior work and discuss their relevance to our approach, highlighting commonalities and differences in focus and methodology.

**Generative models** One natural route to reweighting is to estimate the underlying data density with a generative model and derive weights from that estimate. For a pair of probability measures one can employ generalized energy-based models (Arbel et al., 2020). If one of the measures is the Lebesgue measure, then in principle any density estimation model can be used, including EBMs and both discrete and continuous normalizing flows (Rezende and Mohamed, 2015; Tabak and Turner, 2013; Tabak and Vanden-Eijnden, 2010). In practice, however, accurate density estimation in high dimensions is costly, and more accurate models typically require greater complexity and heavier computation. Thus, we aim to avoid relying on generative density estimation, developing a computationally efficient reweighting.

**Dataset compression and distillation** Compression methods construct a small subset or a synthetic set intended to preserve key statistical properties of the original data (Broadbent et al., 2025; Shetty et al., 2021). Distillation methods, in turn, learn a small

synthetic set, but one that preserves model performance to the same level as training on the full dataset (Cazenavette et al., 2023; Yu et al., 2023). In contrast, our method is model-free and does not aim to retain statistics beyond the support itself. Instead, we focus on adjusting probability mass so that the support is covered more evenly.

**Dataset pruning and filtration**  Unlike distillation, dataset pruning chooses a subset of real examples that minimizes some objective, for example the expected validation loss. Data filtration and cleaning are related tasks that remove low-quality or outlier points from training data (Tan et al., 2023; 2025). These approaches, like compression methods, are usually model-dependent, whereas our method is not. In our framework the central object is a weight function that rebalances the entire distribution. Once this weight function is learned, it can be applied in different ways. For instance, examples can be sampled with probabilities proportional to their weights to form a reduced dataset, or examples with very low weights can be discarded. Thus, pruning arises as one practical downstream use of a more general reweighting principle Additionally, while pruning methods operate with samples, we follow the measure-level viewpoint which includes in particular finite sample setup.

To conclude, we are not aware of any prior work that pursues distribution uniformization through a change of measure while also avoiding explicit density estimation altogether, but our framework fills this gap. In what follows we show that our method is computationally simpler than generative approaches, broader in scope than compression, and naturally compatible with pruning and filtering when applied to empirical datasets.

## 3 PRELIMINARIES

Let $(\Omega, \mathcal{F}, \mathbb{P})$ be a probability space with sample space $\Omega$, $\sigma$-algebra $\mathcal{F}$, and probability measure $\mathbb{P}$ defined on $\mathcal{F}$. Consider random vectors $X : \Omega \to \mathcal{X}$ and $Y : \Omega \to \mathcal{Y}$ with joint distribution $\mathbb{P}_{X,Y}$ and marginals $\mathbb{P}_X$ and $\mathbb{P}_Y$, respectively. We denote product measures by $\mathbb{P}_X \otimes \mathbb{P}_Y$. Wherever needed, we assume the relevant Radon-Nikodym derivatives exist. For two probability measures $\mathbb{Q}$ and $\mathbb{P}$ with $\mathbb{Q} \ll \mathbb{P}$, the Kullback-Leibler (KL) divergence is $\mathsf{D}_{\mathsf{KL}}(\mathbb{Q} \,\|\, \mathbb{P}) = \mathbb{E}_{\mathbb{Q}}\left[\log \frac{\mathrm{d}\mathbb{Q}}{\mathrm{d}\mathbb{P}}\right]$, which is non-negative and vanishes if and only if (iff) $\mathbb{P} = \mathbb{Q}$. The mutual information (MI) between $X$ and $Y$ quantifies the divergence between the joint distribution and the product of marginals:

$$\mathsf{I}(X;Y) = \mathbb{E}\log\frac{\mathrm{d}\,\mathbb{P}_{X,Y}}{\mathrm{d}\,\mathbb{P}_X \otimes \mathbb{P}_Y} = \mathsf{D}_{\mathsf{KL}}\big(\mathbb{P}_{X,Y} \,\big\|\, \mathbb{P}_X \otimes \mathbb{P}_Y\big). \tag{1}$$

When $\mathbb{P}_X$ admits a probability density function (PDF) $p_X$ with respect to (w.r.t.) the Lebesgue measure, the differential entropy is defined as $\mathsf{h}(X) = -\mathbb{E}[\log p_X(X)]$, where $\log(\cdot)$ denotes the natural logarithm. Likewise, the joint entropy $\mathsf{h}(X, Y)$ is defined via the joint density $p_{X,Y}(x, y)$, and conditional entropy is $\mathsf{h}(X \,|\, Y) = -\mathbb{E}\left[\log p_{X \,|\, Y}(X \,|\, Y)\right] = -\mathbb{E}_Y\left[\mathbb{E}_{X \,|\, Y} \log p(X \,|\, Y)\right]$. Under the existence of PDFs, MI satisfies the identities

$$\mathsf{I}(X;Y) = \mathsf{h}(X) - \mathsf{h}(X \,|\, Y) = \mathsf{h}(Y) - \mathsf{h}(Y \,|\, X) = \mathsf{h}(X) + \mathsf{h}(Y) - \mathsf{h}(X, Y). \tag{2}$$

Mutual information can also be expressed through numerous variational bounds (Poole et al., 2019). In this work, we are particularly interested in the Nguyen-Wainwright-Jordan representation (Nguyen et al., 2010):

$$\mathsf{I}(X;Y) = \sup_{T:\mathcal{X}\times\mathcal{Y}\to\mathbb{R}} \mathbb{E}\Big[T(X,Y) - e^{T(X',Y)-1}\Big], \tag{3}$$

where $X'$ is defined to be independent of $X$ and identically-distributed, and $T$ is a so-called critic function.

Finally, we say that $X \to Y$ form a Markov kernel w.r.t. measures $\big\{\mathbb{P}^{(\alpha)}\big\}_{\alpha \in \mathcal{A}}$ defined on $(\Omega, \mathcal{F})$ iff $\forall \alpha, \beta \in \mathcal{A}$ we have $\mathbb{P}^{(\alpha)}_{Y \,|\, X} = \mathbb{P}^{(\beta)}_{Y \,|\, X}$. By default, we omit the measures, implying that $X \to Y$ is a Markov kernel w.r.t. all the probabilities in question.

## 4 General Method

This section derives our InfoMax-based method for dataset distillation and balancing. We begin by discussing the general problem of reweighting a probability measure to achieve a balanced distribution. We then highlight the limitations of a direct balancing approach via KL minimization. For our specific case, we demonstrate that this KL minimization is equivalent to entropy maximization, which we proxy by maximizing the mutual information between original and corrupted data. This connection yields a practical and effective loss function for our task.

### 4.1 Change of Measure and Uniformization

Let $\mathbb{Q}$ be a probability measure on $\mathcal{X} \subseteq \mathbb{R}^d$ with support $S \subseteq \mathcal{X}$ of finite Lebesgue measure $\mu(S) < \infty$. We define another probability measure $\mathbb{P}$ by reweighting $\mathbb{Q}$ with a non-negative Radon-Nikodym derivative (density ratio) $w$:

$$\frac{\mathrm{d}\,\mathbb{P}}{\mathrm{d}\,\mathbb{Q}}(x) = w(x), \quad \mathbb{E}_{\mathbb{Q}}[w(X)] = 1 \tag{4}$$

This relation between $\mathbb{P}$ and $\mathbb{Q}$ establishes corresponding equivalences between expectations:

**Proposition 4.1.** Let $(\mathrm{d}\,\mathbb{P}_X / \mathrm{d}\,\mathbb{Q}_X)(x) = w(x)$. Let $X \to Y$ be a Markov kernel. Then for any measurable $g : \mathcal{X} \times \mathcal{Y} \to \mathbb{R}$ the following holds:

$$\mathbb{E}_{X,Y \sim \mathbb{P}_{X,Y}} g(X,Y) = \mathbb{E}_{X,Y \sim \mathbb{Q}_{X,Y}}[w(X) \cdot g(X,Y)]$$

In this work, we are particularly interested in $\mathbb{P}$ that is as uniform as possible, covering every corner of $S$ with equal probability. One of the natural ways to perform uniformization is to do an information projection, i.e., to minimize the KL divergence between the proposal $\mathbb{P}$ and a uniform distribution $\mathbb{U}(S)$ with support $S$:

$$\mathsf{D}_{\mathsf{KL}}(\mathbb{P} \,\|\, \mathbb{U}(S)) \stackrel{\text{def}}{=} \mathbb{E}_{\mathbb{P}} \log\left(\frac{\mathrm{d}\,\mathbb{P}}{\mathrm{d}\,\mathbb{U}(S)}\right) = \mathbb{E}_{X \sim \mathbb{Q}}\, w(X)\left[\log\left(\frac{\mathrm{d}\,\mathbb{Q}}{\mathrm{d}\,\mathbb{U}(S)}(X)\right) + \log w(X)\right] \to \min$$

Note, however, that minimizing this objective is equivalent to estimating $\mathrm{d}\,\mathbb{Q} / \mathrm{d}\,\mathbb{U}(S)$, which is extremely difficult, especially for complex $\mathbb{Q}$ and unknown $S$.

### 4.2 Entropy Maximization

To avoid density and support estimation altogether, we reframe KL minimization as entropy maximization. This is possible due to the following result:

**Lemma 4.2.** Let $\mathbb{P}$ be a continuous distribution supported on $S$ of finite Lebesgue measure $\mu(S) < \infty$, $\mathbb{U}(S)$ be a uniform probability measure on $S$. Then

$$\mathsf{h}(\mathbb{P}) = \mathsf{h}(\mathbb{U}(S)) - \mathsf{D}_{\mathsf{KL}}(\mathbb{P} \,\|\, \mathbb{U}(S)) \leq \mathsf{h}(\mathbb{U}(S)) = \log \mu(S)$$

*Proof of Lemma 4.2.* Denoting PDF of $\mathbb{P}$ by $p$, we have

$$\mathsf{D}_{\mathsf{KL}}(\mathbb{P} \,\|\, \mathbb{U}(S)) = \int_S p(x) \log\left(\frac{p(x)}{1/\mu(S)}\right) \mathrm{d}x = \int_S \log \mu(S)\, \mathrm{d}x + \int_S p(x) \log p(x)\, \mathrm{d}x$$
$$= \log \mu(S) - \mathsf{h}(\mathbb{P}) = \mathsf{h}(\mathbb{U}(S)) - \mathsf{h}(\mathbb{P})$$

$$\square$$

Therefore, $\mathsf{D}_{\mathsf{KL}}(\mathbb{P} \,\|\, \mathbb{U}(S)) \to \min$ is equivalent to $\mathsf{h}(\mathbb{P}) \to \max$. While this reformulation is helpful, directly maximizing the differential entropy $\mathsf{h}(\mathbb{P})$ still requires some form of density estimation (as differential entropy is defined through PDF), even though the explicit dependence on $S$ has been removed.

To simplify the task enough for practical applications, we leverage (2) to connect mutual information and entropy. Consider a Markov kernel $X \to Y$ w.r.t. a set of measures $\{\mathbb{P}^{(\mu)}\}$ parametrized by $\mu$ according to (4). Recall that $\mathsf{I}(X;Y) = \mathsf{h}(Y) - \mathsf{h}(Y \mid X)$, with the second term being independent of $\mu$ by the Markov kernel definition. Generally, $\mathsf{h}(Y)$ and $\mathsf{h}(X)$ are not related. However, choosing $Y = X + Z$ for a small independent $Z$ yields distribution of $Y$ which is close to that of $X$. Thus, under some mild conditions, maximizing $\mathsf{I}(X;Y)$ also maximizes $\mathsf{h}(X)$ up to a certain additive gap.

**Lemma 4.3.** Let $\mathcal{W}$ be a family of log-Lipschitz continuous functions on a compact set $S$ with $\mu(S) < \infty$, that is, $\mathcal{W} = \{f : \|\nabla \log f(x)\| \leq L,\ x \in S\}$. Let $q(x)$ be a probability density supported on $S$ and define a probability density $p(x) = w(x) \cdot q(x)$ with $w, q \in \mathcal{W}$. Consider independent $Z \sim \mathcal{N}(0, \sigma^2 \mathrm{I})$ and $X \sim p$. Then

$$\mathsf{h}(X + Z) - \frac{\sigma^2 L^2}{2} \leq \mathsf{h}(X) \leq \mathsf{h}(X + Z)$$

The condition $q \in \mathcal{W}$ can be dropped if $\exists w^\star \in \mathcal{W} : \mathsf{h}(X) = \mathsf{h}(\mathbb{U}(S))$.

Intuitively, the log-Lipschitz requirement enforces that the log-density (ratio) cannot change too abruptly across the support. It merely excludes pathological, spike-like densities, which are irrelevant for most practical situations.

Rewriting differential entropy in terms of mutual information allows us to employ variational lower bounds instead of direct density estimation. By leveraging the Nguyen-Wainwright-Jordan representation (3) for $\mathsf{I}(X;Y)$, we get

$$\mathsf{h}(Y) = \sup_{T:\Omega\to\mathbb{R}} \mathbb{E}\left[T(X,Y) - e^{T(X',Y)-1}\right] + \mathrm{const} \tag{5}$$

Proposition 4.1 allows us to write (5) as a functional of $w$ and $T$. Combined with Lemma 4.2 and Lemma 4.3, this yields an upper bound on $\mathsf{D}_{\mathsf{KL}}(\mathbb{P}_X \,\|\, \mathbb{U}(S))$, which is free of density and support estimation:

**Theorem 4.4.** Let $\mathbb{Q}$, $\mathbb{P}$ be probability measures, and let $w = \mathrm{d}\,\mathbb{P}_X\,/\,\mathrm{d}\,\mathbb{Q}_X$. For any measurable critic $T$ define the weighted NWJ loss

$$\mathcal{L}_{\mathrm{wNWJ}}[w, T] = -\mathbb{E}\left[w(X)T(X,Y) - w(X')w(X)e^{T(X',Y)-1}\right],$$

where the expectation is taken over $\mathbb{Q}_X \otimes \mathbb{Q}_{X,Y}$. Assume the conditions of Lemma 4.3 hold, and let $Y = X + Z$ with $Z \sim \mathcal{N}(0, \sigma^2 \mathrm{I})$ independent of $X$, then

$$\mathsf{D}_{\mathsf{KL}}(\mathbb{P}_X \,\|\, \mathbb{U}(S)) \leq \mathrm{const} + \mathcal{L}_{\mathrm{wNWJ}}[w, T],$$

where the inequality is tight up to a gap $\sigma^2 L^2 / 2$.

Minimizing $\mathcal{L}_{\mathrm{wNWJ}}$ with respect to weight and critic functions yields $w(x)$ which reweights $\mathbb{Q}_X$ to $\mathbb{P}_X$ that is close to $\mathbb{U}(S)$. Thanks to the change of measures, it is possible to approximate $\mathcal{L}_{\mathrm{wNWJ}}$ via samples from $\mathbb{Q}$, which are available in practice.

### 4.3 Variance Reduction and Choice of Critic

Note that Theorem 4.4 provides a general recipe, as $w(x)$ and $T(x,y)$ come from wide families of functions. However, in (Poole et al., 2019) it is argued that restricting $T$ and replicating $X$ via i.i.d. copies reduces the variance of $\mathcal{L}_{\mathrm{wNWJ}}$ and allows for a stable learning.

To perform replication, we reinterpret $X$ as a tuple of i.i.d. vectors $(X_1, ..., X_K)$, where $X_1$ now corresponds to the data we want to reweigh, and $X_2, ..., X_K$ serve as negative samples which stabilize the learning (note that they can come from arbitrary distribution). Similarly, we replace $X \to Y$ by $X_i \to Y_i$. This does not affect our previous derivations, as $\mathsf{I}(X_1, ..., X_K; Y_i) = \mathsf{I}(X_i; Y_i)$ due to $X_1, ..., X_K$ being independent.

Next, we restrict $T$ to a family of softmax-based critics which utilize replicated samples. This yields the following InfoNCE-like (Oord et al., 2018) loss function:

**Lemma 4.5.** Under the assumption of Theorem 4.4 define the weighted InfoNCE loss

$$\mathcal{L}_{\mathrm{wNCE}}[w, T] = -\mathbb{E}\left[\frac{1}{K}\sum_{i=1}^{K} w(X_i) \log\left(\frac{e^{T(X_i, Y_i)}}{\sum_{j=1}^{K} w(X_j)e^{T(X_j, Y_i)}}\right)\right],$$

where the expectation is taken w.r.t. $\mathbb{Q}_X^{\otimes K} \otimes \mathbb{P}_{Y\,|\,X}^{\otimes K}$. Then the following bound holds

$$\mathsf{D}_{\mathsf{KL}}(\mathbb{P}_X \,\|\, \mathbb{U}(S)) \leq \mathrm{const} + \mathcal{L}_{\mathrm{wNCE}}[w, T].$$

## 5 EXPERIMENTS

We previously derived fully weighted, theoretically sound estimator for the max-information objective, but in practice its direct optimization can be unstable due to feedback between the learned weights and the critic and scale drift in the softmax. Although Lemma 4.5 is practical under synthetic data (see Section 5.2), to ensure stable, comparable gradients while preserving InfoNCE semantics, we use a practical normalized variant:

$$\mathcal{L}_{\mathrm{mNCE}}[w, T] := \mathbb{E}\left[\frac{1}{K}\sum_{i=1}^{K} w(X_i) \log\frac{e^{T(X_i, Y_i)}}{\sum_{j\neq i} \tilde{w}(X_j)e^{T(X_j, Y_i)} - e^{T(X_i, Y_i)}}\right], \quad (6)$$

where $\tilde{w}(X_j) = \frac{w(X_j)}{\frac{1}{K-1}\left(\sum_{k=1}^{K} w(X_k) - w(X_i)\right)}$.

This choice has three concrete benefits. First, unit positive weight (for $j = i$, $\tilde{w}(X_j) := 1$) preserves the standard InfoNCE numerator/denominator semantics and prevents a degenerate solution in which the optimizer shrinks the loss by inflating the positive's weight. It also leaves the bound interpretation closest to the unweighted case. Next, negative renormalization implements self-normalized importance sampling. It rescales the denominator to the same order as vanilla InfoNCE, making gradients effectively batch-size invariant and keeping temperatures/learning rates transferable across settings, while still changing the relative contribution of negatives according to $w$. Finally, the loss reduces to standard InfoNCE when $w(\cdot) \equiv 1$, ensuring a clean fallback and ruling out exploding/vanishing denominators, when weights become peaky or flat.

### 5.1 IMPLEMENTATION DETAILS

**Model architecture**. We learn (i) a critic $T(x, y)$ for the max-information objective and (ii) an importance scorer $w(x)$ that outputs $\log w(x)$. Both share a small MLP trunk with LayerNorm and a residual block, then branch into two heads: a cosine critic head (two linear layers to 64-d, then cosine similarity; any bounded bilinear critic would work) and a LogRatio head (MLP 64→64→1) producing $f(x)$ with $w(x) = \exp(f(x))$. Please see Listing 1 and Table 4 for details.

**Embedding**. For the synthetic data we operate directly in 2D, without a pre-encoder. For higher-dimensional datasets we use fixed, pretrained self-supervised encoders to obtain latent embeddings on which we learn importance weights: CIFAR-10 uses a ResNet-18 from VICReg (Bardes et al., 2021), output dimension 512; ImageNet-100 uses DINOv2 ViT-Small (Oquab et al., 2023), output dimension 384. Encoders are frozen and labels are never used to train the weight model.

**Objective and stabilizers**. We optimize loss given in Equation (6) with negatives-only normalization (leave-one-out) and no label use. The critic scores $T(\cdot, \cdot)$ appear in the InfoNCE term with temperature $\tau$; the log-weights $f(x)$ reweight samples inside the objective. For stability we use: a smooth cap on log weights (e.g. soft clip to $[-c, +c]$), mean-one normalization of weights per batch.

**Training hyperparameters**. Unless stated otherwise: batch size $\in \{512, 1024, 2048, 4096\}$, temperature $\tau$ depends on the inner dimension of the critic and batch size (generally setting

$\tau = 0.1$ worked universally well in our setups), view noise on embeddings $\mathcal{N}(0, \sigma^2 I)$ with $\sigma \in \{0.001, ..., 0.2\}$ depending on the embedding norms, optimizer AdamW with base learning rate $5 \times 10^{-4}$, weight decay $(10^{-5}, 10^{-6})$, cosine schedule with linear warmup for 10 epochs, for 100 epochs and 5 seeds; we report mean $\pm$ 95% CI.

**Corrected draw**. After training, we freeze scorer $f$ and form a corrected dataset by sampling with replacement at a fixed total budget $M$ (=5000 for CIFAR-10 and ImageNet-100) using probabilities $p_i \propto w(x_i)$. Labels are used only to define the target histogram, not for learning the scorer $f$.

## 5.2 BALANCING A YIN–YANG DENSITY

We construct a 2D synthetic dataset supported on the unit disk $\{x \in \mathbb{R}^2 : |x| \leq 1\}$ with a yin–yang density skew: one half of the disk (separated by a smooth cosine curve) is sampled at a higher rate $(1/\rho)$ than the other. The relatively easier domain allows the use unmodified objective as given by Lemma 4.5.

The three-panel visualization (Figure 2) shows that the learned importance function flattens the skewed density and allows to prune dense regions. For additional evidence, we also report disk-aware summary metrics: *circular variance* (higher is more uniform in angle), *equal-area polar-grid coverage* (fraction of non-empty cells; higher is better), and *polar-grid KL* to uniform (lower is better). See summary statistics in Table 1.

Table 1: **Disk-aware synthetic metrics**. Sampling with $w(x)$ moves the empirical distribution toward uniform over area, as indicated by the metrics. "$\Delta$" is after $-$ before at fixed sample size.

| $\rho$ | setting | circular variance ↑ | polar-grid KL ↓ | polar-grid coverage ↑ |
|---|---|---|---|---|
| | default | $0.418$ $\pm 0.009$ | $0.604$ $\pm 0.007$ | $0.679$ $\pm 0.017$ |
| 0.025 | weighted | $0.520$ $\pm 0.033$ | $0.592$ $\pm 0.100$ | $0.897$ $\pm 0.028$ |
| | $\Delta$ | $+0.103$ $\pm 0.036$ | $-0.012$ $\pm 0.100$ | $+0.218$ $\pm 0.028$ |
| | default | $0.447$ $\pm 0.018$ | $0.541$ $\pm 0.005$ | $0.770$ $\pm 0.014$ |
| 0.05 | weighted | $0.777$ $\pm 0.058$ | $0.242$ $\pm 0.038$ | $0.965$ $\pm 0.000$ |
| | $\Delta$ | $+0.330$ $\pm 0.042$ | $-0.299$ $\pm 0.043$ | $+0.196$ $\pm 0.014$ |
| | default | $0.504$ $\pm 0.007$ | $0.430$ $\pm 0.009$ | $0.878$ $\pm 0.003$ |
| 0.1 | weighted | $0.918$ $\pm 0.025$ | $0.130$ $\pm 0.015$ | $0.997$ $\pm 0.003$ |
| | $\Delta$ | $+0.414$ $\pm 0.031$ | $-0.300$ $\pm 0.020$ | $+0.118$ $\pm 0.005$ |
| | default | $0.639$ $\pm 0.005$ | $0.260$ $\pm 0.007$ | $0.969$ $\pm 0.003$ |
| 0.2 | weighted | $0.966$ $\pm 0.024$ | $0.111$ $\pm 0.012$ | $0.999$ $\pm 0.002$ |
| | $\Delta$ | $+0.327$ $\pm 0.020$ | $-0.149$ $\pm 0.017$ | $+0.030$ $\pm 0.003$ |

## 5.3 DISTRIBUTION CORRECTION ON REAL DATASETS.

We evaluate whether the learned, label-free importance function can correct class skew on CIFAR-10 and ImageNet-100. For each dataset, we construct *Moderate* and *Extreme* regimes and compare the default imbalanced training set against a *corrected draw* obtained by sampling with replacement using probabilities proportional to the learned weights, targeting a uniform class distribution. We visualize per-class histograms (Figure 4 and Figure 3) for Extreme and report deviation metrics at fixed sample size (Table 2 and Table 3).

*MaxAbs* and *L1* are simple deviation-from-uniform measures. MaxAbs is the maximum absolute class deviation $\max_c |\hat{p}_c - \frac{1}{C}|$ and L1 is the $\ell_1$ distance to uniform $\sum_c |\hat{p}_c - \frac{1}{C}|$ (note total-variation distance is $\frac{1}{2} \cdot L1$). Both are lower-is-better ($\downarrow$). For a scalar notion of "how many classes are effectively present," we report two effective class counts: $N_{\text{eff}}^H = \exp(H(\hat{p}))$, the Shannon effective number of classes, and $N_{\text{eff}}^S = \frac{1}{\sum_c \hat{p}_c^2}$, the Simpson effective

number of classes. Both are higher-is-better ($\uparrow$) with target value $C$. Finally, *Gini* is the standard Gini coefficient computed over class proportions (inequality across classes), where lower is better ($\downarrow$).

Table 2: **Class-distribution correction: CIFAR-10.** Baseline shows metrics on default random sample drawn with either Moderate or Extreme imbalance.

| Imbalance | Sampling | MaxAbs $\downarrow$ | L1 $\downarrow$ | $N_{\text{eff}}^H \uparrow$ (10) | $N_{\text{eff}}^S \uparrow$ (10) | Gini $\downarrow$ |
|---|---|---|---|---|---|---|
| Moderate | Random | 0.117 | 0.524 | 8.348 | 7.288 | 0.341 |
| | Corrected $\tau = 0.1$ | 0.034 $\pm$ 0.004 | 0.133 $\pm$ 0.020 | 9.857 $\pm$ 0.019 | 9.720 $\pm$ 0.033 | 0.100 $\pm$ 0.007 |
| Extreme | Random | 0.275 | 0.906 | 5.620 | 4.311 | 0.579 |
| | Corrected $\tau = 0.1$ | 0.088 $\pm$ 0.038 | 0.334 $\pm$ 0.091 | 9.231 $\pm$ 0.398 | 8.504 $\pm$ 0.790 | 0.213 $\pm$ 0.050 |

Table 3: **Class-distribution correction: ImageNet-100.**

| Imbalance | Sampling | MaxAbs $\downarrow$ | L1 $\downarrow$ | $N_{\text{eff}}^H \uparrow$ (100) | $N_{\text{eff}}^S \uparrow$ (100) | Gini $\downarrow$ |
|---|---|---|---|---|---|---|
| Moderate | Random | 0.0174 | 0.5376 | 81.925 | 70.838 | 0.3565 |
| | Corrected $\tau = 0.125$ | 0.008 $\pm$ 0.001 | 0.227 $\pm$ 0.016 | 95.779 $\pm$ 0.538 | 92.329 $\pm$ 0.816 | 0.162 $\pm$ 0.009 |
| Extreme | Random | 0.031 | 0.864 | 59.786 | 46.632 | 0.553 |
| | Corrected $\tau = 0.125$ | 0.012 $\pm$ 0.001 | 0.358 $\pm$ 0.013 | 89.411 $\pm$ 0.443 | 83.338 $\pm$ 0.723 | 0.252 $\pm$ 0.008 |

For each dataset we show per-class bar charts comparing the default (unbalanced) draw to the corrected (learned-weights) draw. Classes are ordered from most frequent to least frequent in the default draw so head–tail effects are obvious at a glance. A dashed horizontal line marks the uniform target $\frac{\text{total\_samples}}{C}$. The corrected bars line up closely with this target across the board: over-represented head classes shrink toward the line, while under-represented tail classes grow toward it. We keep the same total number of samples before and after to make heights comparable, and (optionally) add thin confidence bands from a simple multinomial bootstrap to indicate variability. We show the Extreme regime in the main text (most instructive) and include the Modest regime in the appendix; the qualitative pattern is the same but with smaller gaps.

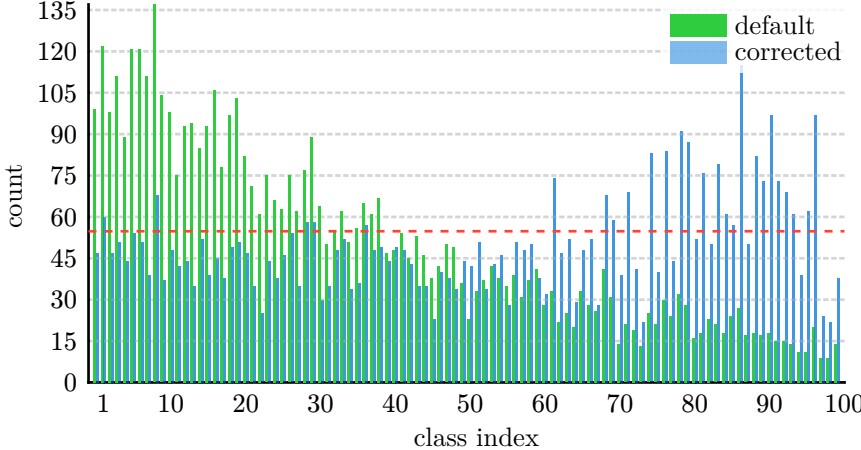

Figure 3: **Per-class proportions (Extreme) on ImageNet-100.** Default (unbalanced) vs corrected (learned weights); dashed line marks the uniform target 50 samples per class.

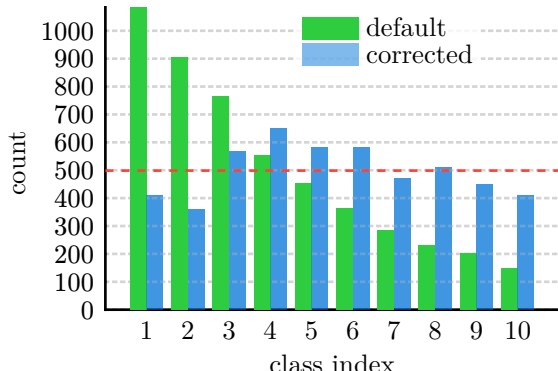

Figure 4: **Per-class proportions (Extreme) on CIFAR-10.** Default (unbalanced) vs corrected (learned weights); dashed line marks the uniform target of 500 samples per class.

## 6 DISCUSSION

**Results.** Information theory often provides universal and rigorous frameworks for various Deep Learning problems. In this paper, we leveraged the connection between dataset redundancy and entropy to propose a novel approach to common data-related problems: balancing, pruning, diversification and redundancy reduction. By pursuing distribution uniformization through entropy maximization, we ensure that underrepresented regions receive appropriate emphasis.

We also note that direct entropy maximization involves explicit support and density estimation, which should be avoided at all costs in high-dimensional cases due to their immense complexity. Therefore, we neatly reframe entropy maximization as mutual information maximization under Gaussian convolutions, which, in turn, allows us to employ cheap and easy-to-implement variational bounds. As a result, we derive a simple contrastive objective and a corresponding reweighting framework. Applying variance-reducing tricks common for contrastive losses further increases practical applicability of our method.

Finally, we validate the proposed approach on synthetic and real tasks. The results indicate that our method is able to detect oversampling and imbalances in both setups. Moreover, we show that it is also possible to use learned weight functions to reduce redundancy via pruning and balance the data via resampling.

**Impact.** This work elaborates on the InfoMax approach proposed in (Bell and Sejnowski, 1995; Linsker, 1988) and refined in (Butakov et al., 2025; Hjelm et al., 2019; Oord et al., 2018) by providing a simple contrastive objective for change of measure tasks while avoiding heavy duty generative modelling.

**Limitations.** In this work, a general framework for distribution uniformization is derived. While it can also be used for dataset balancing and pruning, an additional evaluation is required to conclude whether the proposed approach competes well in these specific tasks.

**Ethics statement.** This work is not subject to any ethical concerns.

**Reproducibility statement.** To ensure reproducibility of our results, we provide complete proofs in Section A and experimental details in Section 5. We also provide our PyTorch implementations of loss functions in the supplementary material.

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

## A  COMPLETE PROOFS

*Proof of Proposition 4.1.*

$$\begin{aligned}
\mathbb{E}_{X,Y\sim\mathbb{P}_{X,Y}}\,g(X,Y) &= \mathbb{E}_{X\sim\mathbb{P}_X}\,\mathbb{E}_{Y\sim\mathbb{P}_{Y\,|\,X}}\,g(X,Y) && \text{(chain rule)}\\
&= \mathbb{E}_{X\sim\mathbb{P}_X}\,\mathbb{E}_{Y\sim\mathbb{Q}_{Y\,|\,X}}\,g(X,Y) && (\,\mathbb{Q}_{Y\,|\,X}=\mathbb{P}_{Y\,|\,X}\,)\\
&= \mathbb{E}_{X\sim\mathbb{Q}_X}\,w(X)\cdot\Big[\mathbb{E}_{Y\sim\mathbb{Q}_{Y\,|\,X}}\,g(X,Y)\Big] && \text{(density def.)}\\
&= \mathbb{E}_{X,Y\sim\mathbb{Q}_{X,Y}}[w(X)\cdot g(X,Y)] && \text{(chain rule)}
\end{aligned}$$

$\square$

*Proof of Lemma 4.3.* We start with the last statement. By Lemma 4.2 we know that $\mathsf{h}(X)=\mathsf{h}(\mathbb{U}(S))$ implies $w(x)=\frac{1/\mu(S)}{q(x)}$, which leads to

$$\|\nabla\log q(x)\| = \|\nabla\log w(x)\| \le L. \tag{7}$$

By De Bruijn's identity (Cover and Thomas, 2006, Theorem 17.7.2),

$$\frac{\mathrm{d}}{\mathrm{d}t}\mathsf{h}\big(X+\sqrt{t}W\big)[w] = \frac{1}{2}J\big(X+\sqrt{t}W\big),\quad W\sim\mathcal{N}(0,\mathrm{I}),\quad W\!\perp\!\!\!\perp X \tag{8}$$

where $J(X_t)=\mathbb{E}\big[\|\nabla\log p_{X_t}(x)\|_2^2\big]$ refers to Fisher information of $X_t=X+\sqrt{t}W$. Integrating (8) yields

$$\begin{aligned}
0 \le \mathsf{h}(X+Z)[w]-\mathsf{h}(X)[w] &= \frac{1}{2}\int_0^{\sigma^2} J\big(X+\sqrt{t}W\big)\,\mathrm{d}t\\
&\le \frac{1}{2}\int_0^{\sigma^2}\mathbb{E}\big[\|\nabla\log w(X)+\nabla\log q_{X_t}\|_2^2\big]\,\mathrm{d}t \le \frac{\sigma^2 L^2}{2}.
\end{aligned} \tag{9}$$

Here the inequality follows from log-Lipschitz continuity of $w(x)$ and Eq. (7), since every convolution satisfies the same bound.

$\square$

*Proof of Theorem 4.4.* First, consider a general case of a Markov kernel $X\to Y$. We start with writing down the NWJ lower bound for $\mathsf{I}(X;Y)$:

$$\mathsf{I}(X;Y) = \sup_{T:\mathcal{X}\times\mathcal{Y}\to\mathbb{R}}\mathbb{E}_{X',X,Y\sim\mathbb{P}_X\otimes\mathbb{P}_{X,Y}}\Big[T(X,Y)-e^{T(X',Y)-1}\Big]$$

Using Proposition 4.1, we perform change of measures:

$$\mathsf{I}(X;Y) = \sup_{T:\mathcal{X}\times\mathcal{Y}\to\mathbb{R}}\mathbb{E}_{X',X,Y\sim\mathbb{Q}_X\otimes\mathbb{Q}_{X,Y}}\,w(X')w(X)\Big[T(X,Y)-e^{T(X',Y)-1}\Big]$$

Since $X'$ and $(X,Y)$ are independent and $\mathbb{E}\,w(X')=1$,

$$\mathsf{I}(X;Y) = \sup_{T:\mathcal{X}\times\mathcal{Y}\to\mathbb{R}}\mathbb{E}_{X',X,Y\sim\mathbb{Q}_X\otimes\mathbb{Q}_{X,Y}}\Big[w(X)T(X,Y)-w(X')w(X)e^{T(X',Y)-1}\Big] \tag{10}$$

By the definition of mutual information (2), we have $\mathsf{I}(X;Y)=\mathsf{h}(Y)-\mathsf{h}(Y\,|\,X)$, with the second term being constant due to $X\to Y$ being a Markov kernel.

Next, we recall that $Y=X+Z$ for an independent $Z\sim\mathcal{N}(0,\sigma^2\mathrm{I})$, and $\mathbb{P}$, $\mathbb{Q}$ and $w$ satisfy the conditions of Lemma 4.3. Therefore, $\mathsf{h}(Y\,|\,X)=\mathsf{h}(X+Z\,|\,X)=\mathsf{h}(Z\,|\,X)=\mathsf{h}(Z)$ and

$$\mathsf{h}(X) \ge \mathsf{I}(X;Y)+\mathsf{h}\big(\mathcal{N}(0,\sigma^2\mathrm{I})\big)-\frac{\sigma^2 L^2}{2}$$

Finally, we employ Lemma 4.2 to derive an upper bound on KL divergence:

$$D_{KL}(\mathbb{P}_X \,\|\, \mathbb{U}(S)) \le \frac{\sigma^2 L^2}{2} + h(\mathbb{U}(S)) - h(\mathcal{N}(0, \sigma^2 I)) - I(X; Y) \tag{11}$$

We finish our proof by substituting (10) into (11):

$$D_{KL}(\mathbb{P}_X \,\|\, \mathbb{U}(S)) \le \underbrace{\frac{\sigma^2 L^2}{2} + h(\mathbb{U}(S)) - h(\mathcal{N}(0, \sigma^2 I))}_{\text{const}}$$

$$-\underbrace{\mathbb{E}\Big[w(X)T(X,Y) - w(X')w(X)e^{T(X',Y)-1}\Big]}_{-\mathcal{L}_{\text{wNWJ}}[w,T]},$$

$\square$

*Proof of Lemma 4.5.* Let $r$ be a uniformly distributed index over the set $\{1, ..., K\}$. By Theorem 4.4, we have an upper bound

$$D_{KL}(\mathbb{P}_X \,\|\, \mathbb{U}(S)) \le \text{const} - \mathbb{E}\Big[w(X_r)\tilde{T}(X_r, Y) - w(X'_r)w(X_r)e^{\tilde{T}(X'_r,Y)-1}\Big],$$

where the expectation now includes averaging w.r.t. random index $r$.

Now consider each term inside the expectation separately, when setting the critic

$$\tilde{T} = 1 + \log\left(\frac{e^{T(X_r,Y)}}{\frac{1}{K}\sum_{j=1}^{K} w(X_j)e^{T(X_j,Y)}}\right).$$

By averaging over the random index $r$ and using linearity of expectation,

$$\mathbb{E}\big[w(X_r)\tilde{T}(X_r, Y)\big] = 1 + \mathbb{E}\left[\sum_{i=1}^{K} w(X_i) \log\left(\frac{e^{T(X_i,Y)}}{\frac{1}{K}\sum_{j=1}^{K} w(X_j)e^{T(X_j,Y)}}\right)\right]$$

$$= 1 + \mathcal{L}_{\text{wNCE}}[w, T].$$

Similarly, taking an expectation over the independent draw $X'_r$, one can simplify the second term as follows

$$\mathbb{E}\big[w(X'_r)w(X_r)e^{\tilde{T}(X'_r,Y)-1}\big]\big] = \mathbb{E}\left[w(X'_r)\frac{e^{T(X'_r,Y)}}{\frac{1}{K}\sum_{j=1}^{K} w(X_j)e^{T(X_j,Y)}}\right]$$

$$= \mathbb{E}\left[\sum_{i=1}^{K} w(X'_i)\frac{e^{T(X_i,Y)}}{\frac{1}{K}\sum_{j=1}^{K} w(X'_j)e^{T(X_j,Y)}}\right] = 1.$$

Substituting the last two equations into the bound on KL divergence above completes the proof. $\square$

# B  IMPLEMENTATION DETAILS

Below is the typical model for critic and importance scorer heads sharing a base network.

```
SharedEncoderT(
  (marginalizer): OuterProductMarginalizer()
  (encoder): Sequential(
    (0): Linear(in_features=384, out_features=64, bias=True)
    (1): LayerNorm((64,), eps=1e-05, elementwise_affine=True)
    (2): GELU(approximate='none')
    (3): ResidualBlock(
      (block): Sequential(
        (0): Linear(in_features=64, out_features=64, bias=True)
        (1): LayerNorm((64,), eps=1e-05, elementwise_affine=True)
        (2): GELU(approximate='none')
        (3): Linear(in_features=64, out_features=64, bias=True)
        (4): LayerNorm((64,), eps=1e-05, elementwise_affine=True)
      )
    )
    (4): Linear(in_features=64, out_features=64, bias=True)
  )
  (critic): BasicCosineCritic(
    (linear_x): Linear(in_features=64, out_features=64, bias=True)
    (linear_y): Linear(in_features=64, out_features=64, bias=True)
  )
  (ratio): LogRatioHead(
    (mlp): Sequential(
      (0): Linear(in_features=64, out_features=64, bias=True)
      (1): LayerNorm((64,), eps=1e-05, elementwise_affine=True)
      (2): GELU(approximate='none')
      (3): ResidualBlock(
        (block): Sequential(
          (0): Linear(in_features=64, out_features=64, bias=True)
          (1): LayerNorm((64,), eps=1e-05, elementwise_affine=True)
          (2): GELU(approximate='none')
          (3): Linear(in_features=64, out_features=64, bias=True)
          (4): LayerNorm((64,), eps=1e-05, elementwise_affine=True)
        )
      )
      (4): Linear(in_features=64, out_features=1, bias=True)
    )
  )
)
```

Listing 1: Example shared critic plus importance scorer model.

Table 4: **Hyperparameters and grids.** Defaults used unless otherwise noted. Ablated on CIFAR-10 Modest imbalance setup, then reused as stable defaults elsewhere.

| Component | Hyperparameter | Default | Grid / Values | Notes |
|---|---|---|---|---|
| Encoders | CIFAR-10 | VICReg ResNet-18 | — | Frozen; 512-d embeddings |
| Encoders | ImageNet-100 | DINOv2 ViT-S | — | Frozen; 384-d embeddings |
| Model | Shared trunk dim | 64 | 32, 64, 128, 256, 512 | Linear d→64, LN, GELU, Residual(64), Linear 64→64 |
| Model | Critic head | Cosine | — | Two linear maps to 64-d; cosine similarity |
| Model | Importance head | MLP 64→64→1 | — | Outputs log-weight f(x) |
| Objective | Temperature $\tau$ | 0.10 | 0.05, 0.10, 0.20, 0.40 | InfoNCE-style |
| Objective | Normalizer | Negatives-only (LOO) | Full SNIS (abl.) | Leave-one-out denominator |
| Training | Batch size B | 2048 | 512, 1024, 2048, 4096 | Per experiment |
| Training | Epochs | — | — | 3 seeds; report mean $\pm$ 95% CI |
| Training | View noise $\sigma$ | — | 0.01, 0.05, 0.10, 0.20 | Gaussian on embeddings |
| Sampling | Budget M | 2048, 5000 | — | 2048 sample for yin yang, 5000 for CIFAR-10/ImageNet-100 |

### B.1 Accuracy in *long-tailed* recognition on CIFAR-10-LT:

We follow the standard CIFAR-10-LT protocol, constructing a synthetic long-tailed training set by subsampling each class so that class sizes decay exponentially from 5000 head samples to 5000/IF tail samples, where IF is the imbalance factor. A pretrained VICReg encoder is kept frozen and used to extract embeddings, on top of which we train a linear classifier with cross-entropy for 10 epochs. For each setting we report the mean and standard deviation over 10 random seeds. In the class balanced (oracle) baseline, we use the true class frequencies to define inverse-frequency class weights in a weighted cross-entropy loss. In the weighted (ours) setting, we instead use our learned weighting model to assign per-sample weights, which are applied in the same weighted cross-entropy objective.

As seen in Table 5 and Figure 5, our uniformization-driven weights (weighted) consistently improve downstream performance over the unweighted baseline (unbalanced), with gains of +0.9, +1.9, +3.1 and +4.9 points as the imbalance factor increases from 10 to 50. As expected, the oracle class-balanced baseline with access to class labels performs best, but our label-free method recovers a substantial fraction of the gap between the unbalanced and oracle settings, especially with growing imbalance.

Table 5: Top-1 accuracy of linear classifier on top of frozen encoder using (**unbalanced**) standard cross-entropy on the imbalanced training set, (**weighted**) our InfoMax-based, label-free per-sample weights, and (**class balanced**) an oracle class-balanced baseline that uses the true synthetic class frequencies to construct inverse-frequency class weights.

| Imbalance Factor | 10 | 20 | 30 | 50 |
|---|---|---|---|---|
| unbalanced | $85.05 \pm 0.08$ | $82.37 \pm 0.15$ | $79.94 \pm 0.42$ | $75.44 \pm 0.84$ |
| **weighted (ours)** | $85.92 \pm 0.13$ | $84.28 \pm 0.21$ | $83.08 \pm 0.11$ | $80.34 \pm 0.15$ |
| class balanced (oracle) | $87.04 \pm 0.06$ | $86.79 \pm 0.13$ | $86.49 \pm 0.15$ | $86.34 \pm 0.14$ |

(a) Imbalance factor 10

(b) Imbalance factor 20

(c) Imbalance factor 30

(d) Imbalance factor 50

Figure 5: Top-1 accuracy over 10 training epochs on long-tailed CIFAR-10 with imbalance factors 10, 20, 30, and 50, comparing unweighted cross-entropy (blue), our InfoMax-based per-sample weighting (red), and the oracle inverse-frequency class-weighted baseline (purple).

