# OpenReview forum: "InfoMax-based Resampling for Dataset Balance and Diversity"
_ICLR.cc/2026/Conference — Submitted to ICLR 2026_

### Official Review · Reviewer_h9Xb · 2025-10-30

**Soundness:** 3
**Presentation:** 2
**Contribution:** 3
**Rating:** 4
**Confidence:** 4

**Summary:**

The paper used implicit differential entropy maximization and developed  a reweighting framework that moves empirical data toward uniform coverage. The authors  exploited intractable entropy maximization rather than a mutual information proxy. The authors also conducted a series of experiments, which showed the usefulness of the proposed framework.

**Strengths:**

1.	This paper explores the resampling problem from an information-theoretic perspective, and the proposed method appears novel.

2.	The motivation is clear — the idea of making the dataset more uniform through entropy maximization appears logical and well-founded.

**Weaknesses:**

1.	In line 58, the method in this paper is not entirely model-free. The InfoMax-based Resampling (IBR) still requires training a model for sampling.

2.	Regarding the resampling experiments in Figures 3 and 4: in large-scale long-tailed scenarios, we do indeed need such data sampling methods. If a uniform distribution could truly be achieved, it would be astonishing. However, the experiments should not use sampling with replacement. A likely situation is that the same tail-class sample was repeatedly selected many times, while the overall number of distinct tail-class samples did not actually increase. In other words, there may be a lack of intra-class diversity. The data shown in Figures 3 and 4 only indicate that many tail-class samples were drawn, but do not demonstrate overall diversity.

3.	Unsupervised learning is intended for downstream tasks. The effectiveness of sampling should not be evaluated solely by the data distribution, but also by whether it improves downstream task performance. The authors lack experiments on additional downstream tasks.

4.	The code is not fully open-sourced; only part of it is available. I tried reproducing the results using the hyperparameters and network settings provided in the paper, but was unable to replicate the results.

**Questions:**

1.	The authors should include an additional deduplicated sampling experiment. We hope that the method can achieve inter-class balance while also maintaining intra-class diversity. Pls give the effective number of resample dataset.

2.	The authors should add further experiments where a model is trained separately on the original dataset and on the resampled dataset. This would allow for a comparison to see whether the long-tail bias of the model has been mitigated, while ensuring that the overall performance remains stable.

3.	I attempted to reproduce the authors’ codes but was unable to replicate the results. It would be good to provide the complete implementation. Moreover, due to  the highly stochastic nature for such sampling experiments,   it might also be more convincing  to provide full codes  to ensure that the reported results reflect true uniformity rather than a random chance.  Overall,the paper looked to me  an piece of interesting  work in the first instance; yet I am a bit disappointed with the code.

---

### Official Review · Reviewer_Hvod · 2025-10-31

**Soundness:** 3
**Presentation:** 2
**Contribution:** 3
**Rating:** 4
**Confidence:** 2

**Summary:**

This paper discusses the problem of dataset balance adjustment. The proposed method replaces entropy maximization with a surrogate measure based on mutual information and derives a variational estimator under measure transformation. The theoretical equivalence of the derivation is demonstrated, and the effectiveness of the approach is validated through experimental results.

**Strengths:**

The proposed method is theoretically well-founded and successfully avoids the need for explicit support or density estimation, which enhances its practical utility. Moreover, the experimental results demonstrate its effectiveness.

**Weaknesses:**

The main concern with this paper lies in the lack of clarity regarding the authors’ central claims and the evidence provided to support them.
The paper primarily focuses on presenting the proposed method and discussing its theoretical derivation. From this perspective, several possible interpretations of the intended claim can be considered:

- **(1) Theoretical compensation as the main contribution:**
    If the primary claim is that the paper provides theoretical compensation or justification for the proposed method, it should be made clear whether the conceptual foundation of the method had already been acknowledged in prior work but lacked a concrete formulation, or whether its utility has been newly demonstrated. In either case, the novelty and necessity of the theoretical contribution should be explicitly articulated.

- **(2) Avoiding explicit support or density estimation as the key claim:**
    If the emphasis lies on the practical advantage of avoiding explicit support or density estimation, then the theoretical discussion mainly serves as a derivation. In that case, the experiments should directly correspond to this claim—for instance, by comparing the performance differences between conventional approaches that rely on support or density estimation and the proposed method.

- **(3) Superior performance for dataset balance adjustment as the key claim:**
    If the paper claims that the proposed method provides a superior solution to the problem of dataset balance adjustment, then comparisons with state-of-the-art methods addressing similar tasks are necessary to validate the claimed advantage.


In summary, the paper would benefit from clarifying which of these claims constitutes its main contribution and aligning the theoretical discussion and experiments accordingly.

**Questions:**

As mentioned in the _Weaknesses_ section, the main claim of the paper should be clearly stated. Please clarify what the central contribution or assertion of the work is, and explicitly describe how the evidence presented in the paper—both the theoretical proofs and the experimental results—supports that claim. Establishing this connection between the stated objective and the provided evidence would greatly improve the clarity and coherence of the paper.

---

### Official Review · Reviewer_VVjZ · 2025-10-31

**Soundness:** 2
**Presentation:** 2
**Contribution:** 2
**Rating:** 2
**Confidence:** 3

**Summary:**

This paper proposes an information-theoretic approach for addressing data imbalance through distribution uniformization. The authors formulate a weighting-based objective to learn sample weights for balancing, and validate the approach on both synthetic and real datasets.

**Strengths:**

1. The proposed method is motivated by information-theoretic principles and supported by theoretical analysis.

**Weaknesses:**

1. The paper is not well organized or clearly written. In particular, Section 4 presents multiple theoretical results, but it is not clear which part constitutes the proposed method. Please see Questions below for details.
2. I have concerns regarding the significance and novelty of the study. The relationship between this work and prior research is not clearly established.
3. The paper does not provide sufficient details about the simulation experiments, such as the runtime or computational setup.

**Questions:**

1. Section 4 presents multiple theoretical results, but it is not clear which part constitutes the proposed method.
2. It is not clear how the loss function in Lemma 4.5 relates to the loss defined in Equation (6).

---

### Official Review · Reviewer_Zv9T · 2025-10-31

**Soundness:** 3
**Presentation:** 3
**Contribution:** 2
**Rating:** 4
**Confidence:** 3

**Summary:**

The paper proposes a principled reweighting framework that aims to move empirical data toward uniform coverage through implicit differential entropy maximization. The core idea replaces intractable entropy maximization with a mutual information proxy and derives variational estimators under change of measure, yielding a consistent, low-variance weighted InfoNCE-based objective. The learned weights can be used directly for data filtration and imbalance-aware sampling, without requiring explicit estimation of the data manifold or high-dimensional densities.

**Strengths:**

1. Does not require labels or generative modeling; only relies on pretrained embeddings and contrastive losses.

2. Weights can be reused for various downstream tasks.

3. Addresses redundancy and imbalance in modern large-scale datasets, a key open problem for efficient ML training.

4. Quantitative and qualitative improvements shown across synthetic and real datasets.

5. Effective across imbalance regimes (moderate/extreme).

6. Provides full proofs, implementation details

**Weaknesses:**

1. The paper contains limited comparison with baselines, since there are no empirical comparison to existing data-balancing or reweighting methods (e.g., focal loss, effective number weighting, re-sampling baselines).

2. The results show improvement but without contextual baselines, significance is unclear.

3. The paper shows improved distribution uniformity but doesn’t test whether this leads to better model performance (accuracy, robustness, etc.).

4. Although the authors argue stability via normalization, the loss could still be sensitive to extreme weights or embeddings with non-uniform norms, more analysis should be here.

5. This method works well on embeddings, but unclear if it can be used directly on raw data or in streaming scenarios.

**Questions:**

1. How does InfoMax-based reweighting compare with other baselines?

2. Does improved distribution uniformity translate to better test accuracy or generalization?

3. How robust is the learned weighting to encoder choice, temperature, and sigma?

---

### Author Response · Authors · 2025-12-03
**Final comment**

Dear Reviewers,

Thank you for your time and effort in reviewing our work. We apologize for not being able to engage in individual discussions prior to the discussion shutdown due to the unforeseen consequences. As individual communication is no longer feasible, we are providing this unified response.

We are pleased that the reviewers found our work theoretically grounded. We agree with the key concerns raised, particularly regarding the lack of downstream task evaluation and comparisons to other methods. Motivated by the reviewers' valuable feedback and thorough comments, we have begun a major revision of the manuscript. We sincerely appreciate the time spent evaluating our work and assure you that all suggestions will be incorporated into our next submission.

To at least partially address the experimental concerns in the interim, we are providing new results demonstrating that our uniformization approach improves downstream performance. Specifically, we added a set of downstream classification experiments on long-tailed CIFAR-10. In this setting, we train a linear classifier on a frozen encoder (VICReg) using (a) standard cross-entropy on the imbalanced training set (unbalanced), (b) our InfoMax-based, label-free per-sample weights (weighted (ours)), and (c) an oracle class-balanced baseline that uses the true synthetic class frequencies to construct inverse-frequency class weights (class balanced). We evaluate balanced accuracy on a balanced validation set and report mean ± standard deviation over 10 random runs. As shown in Table 5 and Figure 5 (Appendix B.1) of the rebuttal revision, our method halves the accuracy gap between the unbalanced and balanced settings, thereby improving final accuracy, especially in the more strongly imbalanced regimes. We also recapitulate Table 5 below:

|Imbalance Factor|$10$|$20$|$30$|$50$|
|--|--|--|--|--|
|Unbalanced|$85.05 \pm 0.08$|$82.37 \pm 0.15$|$79.94 \pm 0.42$|$75.44 \pm 0.84$|
|Weighted **(ours)**|$85.92 \pm 0.13$|$84.28 \pm 0.21$|$83.08 \pm 0.11$|$80.34 \pm 0.15$|
|Class balanced (oracle)|$87.04 \pm 0.06$|$86.79 \pm 0.13$|$86.49 \pm 0.15$|$86.34 \pm 0.14$|

We would also like to further contextualize our work within the existing literature. As noted in the introduction, the informal concept of using a maximum entropy principle for data rebalancing has indeed been discussed previously, both explicitly and implicitly. Our core contributions are to: (1) provide a formal theoretical grounding for this intuition, (2) establish a connection between the maximum entropy and maximum mutual information objectives, and (3) introduce an efficient contrastive proxy that is straightforward to optimize.

Once again, we thank all the reviewers for their work and insightful critique.

---

### Meta-Review · Area_Chair_eXwa · 2026-01-06

**Summary:**

The reviewers agreed that this paper has some nice ideas and presents a theoretically grounded approach to data rebalancing. However, the paper lacks strong comparisons with existing methods and evaluation of the downstream performance of the data rebalancing method.  Additionally, the main contributions/key takeaways of the paper need to be more clearly articulated/laid out. The authors agree that these are important issues and plan to revise and resubmit based on reviewer feedback.

**Reviewer Concerns:**

The authors generally agreed with the reviewer concerns -- they provided some initial experiments addressing the lack of evaluation on downstream tasks. They also briefly clarified the goal of the paper -- to provide a formal foundation for the high-level idea of using maximum entropy for data rebalancing, which has been considered before. In the revision, they should more carefully discuss prior work that uses this idea and explain what they are adding to it.

**Reviewer Scores:**

I don't think any would have changed. The rebuttal was brief.

---

### Decision · Program_Chairs · 2026-01-26

Reject